# Deep Multi-Class Segmentation Without Ground-Truth Labels

**Thomas Joyce**
School of Engineering
University of Edinburgh
West Mains Rd, Edinburgh EH9 3FB
t.joyce@ed.ac.uk

**Agisilaos Chartsias**
School of Engineering
University of Edinburgh
West Mains Rd, Edinburgh EH9 3FB
agis.chartsias@ed.ac.uk

**Sotirios A. Tsaftaris**
School of Engineering
University of Edinburgh
West Mains Rd, Edinburgh EH9 3FB
s.tsaftaris@ed.ac.uk

## Abstract

In this paper we demonstrate that through the use of adversarial training and additional unsupervised costs it is possible to train a multi-class anatomical segmentation algorithm without any ground-truth labels for the data set to be segmented. Specifically, using labels from a different data set of the same anatomy (although potentially in a different modality) we train a model to synthesise realistic multi-channel label masks from input cardiac images in both CT and MRI, through adversarial learning. However, as is to be expected, generating realistic mask images is not, on its own, sufficient for the segmentation task: the model can use the input image as a source of noise and synthesise highly realistic segmentation masks that do no necessarily correspond spatially to the input. To overcome this, we introduce additional unsupervised costs, and demonstrate that these provide sufficient further guidance to produce good segmentation results. We test our proposed method on both CT and MR data from the multi-modal whole heart segmentation challenge (MM-WHS) [1], and show the effect of our unsupervised costs on improving the segmentation results, in comparison to a variant without them.

## 1 Introduction

Deep learning methods are increasingly being applied in the medical domain, and have demonstrated successes in diverse medical image processing tasks across various anatomies [7]. Here we are interested in the segmentation of cardiac images, which offer particular challenges with the underlying anatomy varying in shape, as typical of an active muscle. Specifically, we focus on the segmentation of the Left Ventricle (LV), Right Ventricle (RV) and Myocardium (MYO) regions of cardiac MR and CT images. Both MR and CT modalities have important clinical applications making automatic segmentation a valuable tool [23]. Deep learning approaches have previously been applied to the cardiac segmentation task, but typically these perform supervised segmentation, and thus require extensive annotated images, which is not always possible because of the difficulty in obtaining the data and the required expertise by the annotators.

In this paper, we present a method for cardiac segmentation which does not require a training set of paired images and ground-truth segmentation labels. Instead, we make use of example labels

1st Conference on Medical Imaging with Deep Learning (MIDL 2018), Amsterdam, The Netherlands.

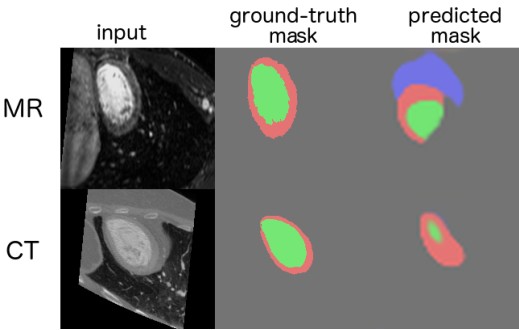

Figure 1: An example of the problem that can arise when training an unsupervised segmentation algorithm using *only* an adversarial loss, such that the only goal is to produce realistic masks. The predicted masks are realistic, but do not correspond to the underlying anatomy.

coming from any previously labelled cardiac data set, i.e. not necessarily from images of the same modality or the same patients as the images of interest.

In order to achieve segmentation, we train a Generative Adversarial Network (GAN) [4] model to synthesise realistic masks from input images. However, as we will demonstrate, minimising only an adversarial cost is not a sufficiently restrictive goal. While the network can produce realistic masks from input images, these masks do *not* necessarily have a pixel-to-pixel correspondence with the underlying substructures in the input image (see Figure 1). However, this is to be expected, as the task is under-restricted: there is no requirement for the mask produced to be *the* mask of the input image. We believe the generator is able to essentially treat the input image as a source of noise, and can then behave like a traditional GAN, and synthesise a realistic output image. To overcome this problem we propose a model with a number of additional unsupervised costs that aims to promote the discovery of regions (defined by the masks) of high similarity. In particular, we encourage intensity similarity in the segmented regions, encourage the segmented regions to be large whilst staying realistic, and introduce an extremely simple reconstruction network to allow a reconstruction cost to be included, without creating the potential for further alignment problems to develop. The joint optimisation of all costs results in masks that are not only realistic but also correspond (spatially) to the input image.

The contributions of the paper are as follows. We demonstrate the possibility for multi-class cardiac segmentation without labels on the data set of interest through adversarial training. We show that adversarial training alone is not sufficient for the unsupervised segmentation task, and we propose a neural network model with an encoder-decoder architecture and a number of unsupervised costs that improve the segmentation performance when used in conjunction with adversarial training. Finally, we perform an ablation study on the proposed costs, showing that the best results are achieved through their combination.

We demonstrate our approach on segmentation of three cardiac regions on both an MR and a CT data set from MM-WHS. We evaluate the accuracy of our results by comparing with an upper bound obtained by training a U-Net [16] with full supervision, and also by comparing with a standard GAN model that does not use our proposed costs.

The paper now proceeds as follows: we first provide an overview of our task in Section 2. We then summarise, in Section 3, related literature in the field of segmentation with or without labels. Section 4 describes in detail our approaches to unsupervised segmentation. In Section 5 we experimentally evaluate our approaches and finally we conclude in Section 6.

## 2   Problem Overview

The problem of segmentation can be seen as a function learning problem. Specifically, an $m$-class 2D image segmentation task can be seen as learning a function $f : \mathbf{R}^{h,w,c} \rightarrow \{0,1\}^{h,w,m}$, where $I \in \mathbf{R}^{h,w,c}$ is an input $h \times w$ pixel $c$ channel input image, and $f(I)$ is an $m$-channel binary image of the same spatial size. Thus, learning a segmentation algorithm is learning a suitable function $f$.

Here we are interested in segmenting CT or MR images into three regions (MYO, LV and RV), so we have $c = 1$ and $m = 3$.

We will represent $f$ as a neural network. Thus, the aim is to specify how $f$ should be trained to produce the right mapping. In the supervised setting (results from which we provide in Section 5.4), $f$ can be trained by minimising the error on known image-mask pairs. In this paper we explore how $f$ can be trained without such paired data.

## 3 Previous Work

Here we review relevant previous work on cardiac image segmentation. We also survey related work on unsupervised segmentation, and segmentation with unlabelled data.

### 3.1 Cardiac Image Segmentation

There has been much previous work on automatic segmentation of anatomy from cardiac images, with state of the art results currently achieved with fully convolutional deep neural networks, such as the 2D fully supervised approach in [18]. Supervised segmentation results can be further improved by considering adjacent volume slices [21] or by introducing shape priors [26, 12]. Multi-class segmentation has also been investigated in the 3D setting, again this can produce improved performance, see for example [13]. Note however, that all of these approaches require extensive labelled training data.

### 3.2 Unsupervised Segmentation

Unsupervised segmentation attempts to overcome the labelled data requirement, and is a more challenging problem. To the best of our knowledge this work is the first deep learning approach to unsupervised cardiac segmentation. That said, there are a small number of previous approaches to unsupervised cardiac segmentation using non deep learning methods. In [3] the myocardium is initially detected by fitting a Gaussian Mixture Model to represent the different tissue characteristics, and then a Markov Random Field (MRF) is optimised based on the likelihood distribution of the intensity and gradient of pixels in the detected region. In [10] a sparse representation is firstly obtained with dictionary learning of a coarse segmentation, and secondly this representation is segmented with a Support Vector Machine pixel classifier. This is extended in [11], in which dictionary learning is combined with a new pre-processing step and Markov Random Fields to further improve segmentation accuracy. However, these approaches only address single-class segmentation and do not tackle the multi-class problem.

Recently, an encoder-decoder architecture for unsupervised semantic segmentation has been proposed in [19] in which the encoder encodes an input image into a multi-class segmentation map that is then decoded to produce the original input. The segmentation map is constrained by a soft cut loss and post-processed by conditional random fields and hierarchical merging of areas. This is most related to our work, since its architecture is also an autoencoder. However, this method is not end-to-end, requires post-processing of the intermediate representation to produce semantically meaningful masks, and also does not use adversarial training.

### 3.3 Other Related Works

More generally, extracting a multi-class semantic mask from an image can be seen as a form of lossy compression, or as a representation learning task. As in the variational lossy autoencoder [2], our aim here can be seen as capturing structural information, and discarding other irrelevant information. The aim is to discard unnecessary information from the input whilst retaining the salient features, which here correspond to the underlying anatomical structure. In this sense the unsupervised segmentation task can be seen as a particular example of the more general unsupervised representation learning problems [15].

# 4 Proposed Approach

We define a segmentation network consisting of a shallow U-Net like architecture with only 2 down-sample / up-sample stages, LeakyReLU activations and Instance Normalisation [17], with a softmax activation on the final layer. This segmentation network will act as the generator in our adversarial training setup, taking either 2D CT or MR cardiac images as input and producing a three-channel segmentation mask as output. Additionally, in parallel we train a discriminator network to be used for the adversarial training of the generator. To improve the performance of the adversarial training we use the Least-Squares GAN (LSGAN) loss-function [8], and employ Spectral Normalisation [9] in the discriminator.

We now describe firstly the initial simple adversarial approach, and then our improved adversarial approach in detail.

## 4.1 Adversarial Approach

Generative adversarial learning [4] is now often used when paired data is unavailable in order to learn image transformations, for example with the use of a cycle consistency property [5, 20, 22], or directly to synthesise realistic data from noise [15]. In this case segmentation can be perceived as a special case of image generation, thus an adversarial loss can be used to train a deep neural network to produce realistic results. As seen in Figure 1 and discussed in Section 5.4, this naive approach does not guarantee that each binary region of the segmentation map is spatially aligned with its corresponding region in the real image, concluding that just an adversarial cost is not sufficient for our task. In particular, although this adversarial approach often produces good synthetic masks, these masks, despite being realistic, are only very roughly related to the underlying image. The results of this approach are given in Section 5.4.

Here, given an input image $X$ we are interested in segmenting MYO, LV and RV, represented as a 3-channel mask $Z_m = \{Z_{MYO}, Z_{LV}, Z_{RV}\}$. Given real three-channel masks $M = \{M_{MYO}, M_{LV}, M_{RV}\}$, our LSGAN based adversarial cost is defined by a discriminator $D$:

$$c_1(X, Z_m, f) = D(M)^2 + (D(Z_m) - 1)^2.$$

Further training details are given in Section 5.2.

## 4.2 Proposed Adversarial Approach

Although training a generator to produce realistic synthetic masks is possible in the above adversarial setup, the resulting images are often not well correlated with the input. In order to overcome this we propose a number of additions to the simple adversarial training.

Firstly, as well as predicting the segmentation mask $Z_m$ we also produce a multi-channel residual $Z_b$, which can store the non-mask information, that is $f(X) = \{Z_m, Z_b\}$. In our work we used a $Z_b$ with 4 channels, but found the exact value didn't have a large influence on the results. $Z_m$ and $Z_b$ are concatenated together to produce a 7-channel latent representation $Z$. Based on this $Z$ we then try to reconstruct the original input as follows: a reconstruction network $h$ inspired by the conditional normalisation in FiLM [14] predicts two 7 element vectors $\gamma$ and $\beta$. The final reconstruction is then simply $\sum_{i=1}^{7} Z_i \gamma_i + \beta_i$, where $Z_i$ is the $i$-th channel of $Z$ and $\gamma_i, \beta_i$ are the $i$-th values in $\gamma$ and $\beta$ respectively. A schematic is given in Figure 2.

Thus, our model functions like an auto-encoder, with the segmentor acting as an encoder, encoding an image $X$ to a mask prediction $Z_m$ and residual information $Z_b$. The reconstructor network $h$ then takes $Z_m$, $Z_b$ and $X$, and following a very simple structure tries to reconstruct $X$ from a weighted sum of the channels of $Z_m$ and $Z_b$.

In additional to the LSGAN based adversarial cost defined above, which we still apply to $Z_m$, we also introduce three additional costs. Firstly an autoencoder like reconstruction loss:

$$c_2(X, f, h) = |X - h(f(X), X)|.$$

Secondly, the produced segmentation masks are encouraged to be large, in order to avoid segmenting sub-regions that still appear realistic:

$$c_3(Z_i) = -\sum Z_m.$$

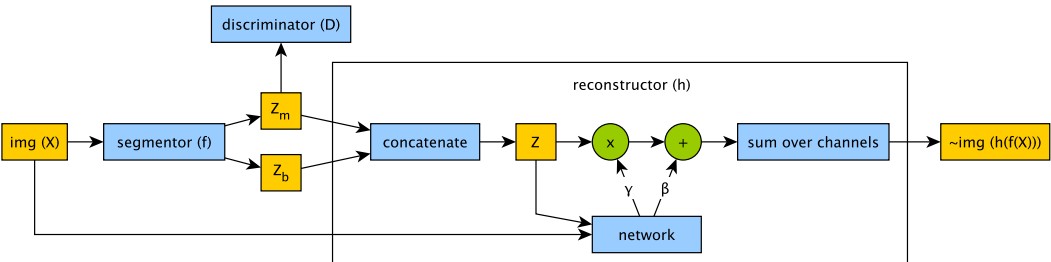

Figure 2: Schematic for our proposed approach. A segmentation network $f$ receives an input image $X$ and produces a multi-channel feature map. The first three channels, $Z_m$, contain segmentations of LV, RV and MYO, as encouraged by a mask discriminator $D$. The residual channels $Z_b$ of the feature map along with $Z_m$ are used as the input to a reconstruction network $h$ that synthesises the input image. The network $h$ is also conditioned on the input image to provide additional information of what intensities to use for each reconstructed region.

Where the sum is over all channels and pixels. Finally, to encourage delineated regions to have similar intensity values, we also minimise the within-region-variance of intensity values:

$$c_4(X, Z) = \sum_i var(X \odot Z_{m,i})$$

where $Z_{m,i}$ denotes the $i$-th channel of the mask $Z_m$, and $\odot$ is the element-wise product. The overall cost function is a weighted sum of the individual costs $C = \lambda_1 c_1 + \lambda_2 c_2 + \lambda_3 c_3 + \lambda_4 c_4$, where $\lambda_1 = \lambda_2 = \lambda_3 = 1$ and $\lambda_4 = 100$. Due to a big difference in the values produced by $c_4$ in comparison with the other costs, a $\lambda_4 = 100$ has been set.

## 5    Experiments

Here we evaluate our approach by generating binary masks of the MYO, LV and RV regions of the heart and compare with an upper bound, as obtained by fully supervised segmentation, and the naive unsupervised segmentation approach described in Section 4.1. In Section 5.1 we describe the data used for our evaluation, Section 5.2 describes the network architecture and training details and finally Section 5.4 describes the experimental results.

### 5.1    Data and Pre-processing

For all experiments we use the 2017 MM-WHS challenge dataset [23, 24, 25], which consists of 20 CT/CTA and 20 MRI volumes. The CT/CTA data were acquired at Shanghai Shuguang Hospital, China, using routine cardiac CT angiography. The slices were acquired in the axial view. The inplane resolution is about $0.78 \times 0.78mm$ and the average slice thickness is $1.60mm$. The MRI data were acquired at St. Thomas hospital and Royal Brompton Hospital, London, UK, using 3D balanced steady state free precession (b-SSFP) sequences, with about $2mm$ acquisition resolution at each direction and reconstructed (resampled) into about $1mm$. The data contains static 3D images, acquired at different time points relative to the systole and diastole. All the data has manual segmentations of the seven whole heart substructures. We removed images that did not contain at least 400 pixels of myocardium, restricting our attention to central slices, as basal and apical slices can be challenging even for supervised approaches and our adversarial training was not stable when all slices were used.

For the unsupervised case we also down-sample the images four times before segmenting, and then up-sample the resulting segmentation mask to compute the Dice. This was done in order to facilitate training of the adversarial networks, which have proven unstable when dealing with larger size images.

Our data is pre-processed as follows: first the field of view is made approximately consistent across the volumes with affine transformations, then images are cropped to a region of interest around the heart. Finally the intensities are normalised to be in the range $[-1, 1]$. This results in 2580 images of size $176 \times 192$ pixels for each modality.

Table 1: Dice score of MYO, LV and RV regions for supervised (top two rows) and unsupervised (bottom four rows) segmentation approaches.

|  |  | MYO | LV | RV | Average |
|---|---|---|---|---|---|
| **Supervised (upper bound)** | MR | 0.76 | 0.90 | 0.87 | 0.84 |
|  | CT | 0.85 | 0.90 | 0.85 | 0.87 |
| **Simple GAN** | MR | 0.42 | 0.66 | 0.55 | 0.54 |
|  | CT | 0.31 | 0.39 | 0.29 | 0.33 |
| **Proposed GAN** | MR | 0.56 | 0.78 | 0.65 | 0.66 |
|  | CT | 0.44 | 0.66 | 0.42 | 0.51 |

## 5.2 Network Architectures and Training

For the supervised baseline we train a standard U-Net model [16], with the final layer changed to a three filter 2D convolution using a sigmoid activation, so that the network outputs the required 3-channel masks. The U-Net consists of 4 convolution and down-sampling blocks, followed by 4 convolution and up-sampling blocks. We train the model using Adam [6] with standard parameters, stopping when no improvement is seen on a validation set. The adversarial networks in the unsupervised settings are trained for a fixed number of 100 epochs. The unsupervised MR segmentation model is trained with segmentation masks from the CT dataset and vice versa, in order to avoid the possibility of the network memorising masks and learning to match memorised masks to images.

For all experiments we use 3-fold cross validation, splitting the data into a 12 volume training set, a 4 volume validation set and a 4 volume test set for each split. The division into each split is random (although fixed across experiments) with the only restriction being that in each split the test set contains different volumes. All models are implemented in Python using Keras.

## 5.3 Experimental Method

We train across 3 splits, repeating each split 7 times. We then take the splits in which the generator successfully learnt to generate all three anatomical regions (which we assessed automatically by only including models that achieved over 10% Dice on the test set for each of the three regions, which we used as a proxy for selecting only models which produced realistic masks). When training the MR model we use the segmentation masks from the CT data as 'real' examples for the discriminator, and vice versa for training on CT images.

## 5.4 Segmentation Results

Here we evaluate our two approaches for unsupervised segmentation and compare with the supervised (upper bound). The Dice scores of the three experiments are summarised in Table 1. Supervised training of a U-Net results in a mean Dice of 0.84 and 0.87 for MR and CT respectively. Training a GAN with our proposed costs of Section 4.2 outperforms the results from a standard GAN in all three regions, producing a mean Dice of 0.66 and 0.51 for MR and CT respectively. Example results from our model using all costs are shown in Figure 3.

## 5.5 Costs evaluation

In this experiment we perform an ablation study to evaluate the effect of the four cost functions described in Section 4.2. Table 2 presents Dice scores in four situations: when using just the adversarial cost $c_1$, when adding the reconstruction cost $c_2$, when combining $c_1$ with maximising the size of the mask $c_3$ and minimising the within region pixel variance $c_4$ and finally when using all costs. We observe the results improve when $c_3$ and $c_4$ are included, while the best performance is obtained when all four costs are jointly optimised.

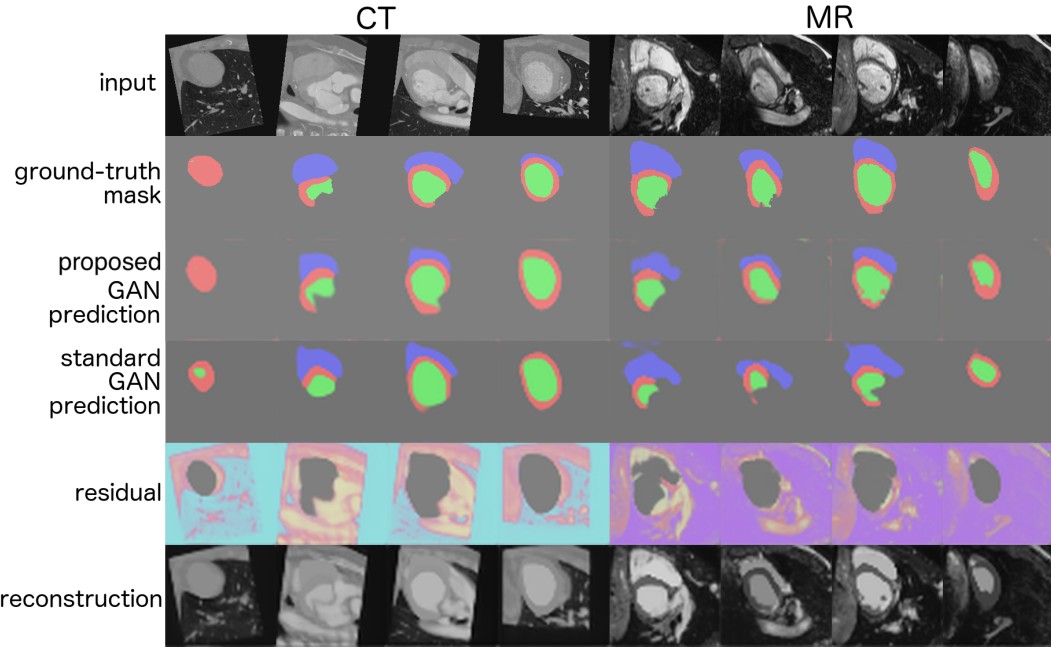

Figure 3: Unsupervised segmentation examples. The first two rows show input images and ground truth segmentation masks (of LV, RV and MYO) respectively. The next two rows show segmentation results from our proposed and a simple adversarial method. Finally, the last two rows show examples of the residual channels and input reconstruction images respectively. As can be seen the unsupervised segmentation is able to capture the anatomical structure, although it has problems with over and under segmentation (see discussion in Section 6). For example, the right ventricle is systematically under segmented in all examples shown, when present. (Note the colours in the residual images differ between the two experiments as the channels in the residual are used differently (for example, are differently ordered) as there is no explicit cost controlling the residual structure.)

Table 2: Ablation study evaluating the effect of different costs. The Dice score of our proposed approach with different cost combinations is reported on the same test volumes.

|  | $c_1$ | $c_1, c_2$ | $c_1, c_3, c_4$ | all costs |
|---|---|---|---|---|
| MR | 0.54 | 0.58 | 0.64 | 0.66 |
| CT | 0.33 | 0.45 | 0.43 | 0.51 |

## 6   Discussion and Conclusion

We have shown that the multi-class segmentation task can be approached even when no labels on the data set of interest are available, demonstrating that an adversarially trained model with suitable costs can produce reasonable results on both MR and CT cardiac data. Further, we demonstrated that an unrestricted adversarial approach led to realistic but erroneous synthetic mask images, essentially treating the input as a source of noise. Although not surprising in itself, this behaviour is important to be aware of when applying machine learning techniques to medical image tasks in limited data settings. We discussed potential approaches to overcoming this 'treating input as noise' problem, in particular demonstrating that additional costs combined with an auto-encoder style approach can suitably restrict the learnt function. Further understanding the relationship between implicit and explicit restrictions and learnt functions is an open and interesting area of machine learning research, with particular relevance in medical imaging, as this is a domain in which accuracy is particularly important, as is properly understanding the learnt behaviour of our models.

We have shown that unsupervised segmentation can sometimes over or under segment a region, since partial or expanded masks can look like realistic masks. However, the model is still achieving broad localisation, and producing promising approximate masks for the underlying multi-class anatomy. A

potential extension could be to expore computing the within-class variance cost $c_4$ in a representation space, rather than directly in the pixel space. This could be done either with features learnt by the segmentor itself, or with an external feature extractor.

Although here we used the masks from a different data set of the same anatomy, it would also be possible to instead use a cardiac shape model to generate realistic mask shapes. This would overcome the need for expert labelling, and could also potentially allow a very large number of example masks to be generated.

**Acknowledgements**

This work was supported in part by the US National Institutes of Health (1R01HL136578-01) and UK EPSRC (EP/P022928/1). We thank NVIDIA for donating a Titan-X GPU.

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
