# OpenReview forum: "Deep Multi-Class Segmentation Without Ground-Truth Labels"
_MIDL.amsterdam/2018/Conference — MIDL 2018 Oral_

### Review · AnonReviewer2 · 2018-05-01
**Review of Deep Multi-Class Segmentation Without Ground-Truth Labels**

**Rating:** 5
**Confidence:** 2

**Review:**

In this paper, the authors proposed unsupervised semantics segmentation with  the use of adversarial training and additional unsupervised costs to train a multi-class anatomical segmentation algorithm. It looks very challenges.
1. Please identify SD of accuracies in Table 1, Table 2.
2. Please shows number of each classes including  LV, RV, MYO in CT or MRI.


**Special Issue:**

Definitely

---

### Review · AnonReviewer3 · 2018-05-09
**Nice approach to regularise GAN training for segmentations but not completely unsupervised**

**Rating:** 3
**Confidence:** 3

**Review:**

Authors present a method for training segmentation network without paired dataset, i.e. images and corresponding ground truth labels. The method is based on training a GAN using ground truth labels coming from a different dataset of the same anatomy. Authors show short-comings of training a GAN for generating segmentations and propose several additional cost functions to regularise it.

Pros:
1. Nice method to regularise GAN training for segmentation related tasks.
2. Evaluation of different parts of the cost function, i.e. ablation study, is well executed and convincing.
3. Results suggest the value of the additional cost terms.
4. Article is well written.
5. Although there is no direct application of the method, the concept is interesting and can be useful for a variety of other studies.
Cons:
1. The method is not unsupervised. Authors use ground-truth labels coming from a different modality but of the same anatomical structure. I think using the term unsupervised segmentation is misleading. Likewise, the title is also misleading. The method cannot work without ground truth labels from the other dataset.
2. To the best of my understanding, the proposed method is addressing the same problem as unsupervised domain adaptation addresses. Yet, there is no discussion of that literature nor any comparison. There has been a couple of recent studies on the topic and it would be very useful to see a comparison and discussion.

Overall, the article has certain problems due to the proximity to unsupervised domain adaptation literature. However, constraining GAN training with the cost functions is interesting and can be useful for various other applications.

**Special Issue:**

No

---

### Review · AnonReviewer1 · 2018-05-09
**An outstanding paper**

**Rating:** 5
**Confidence:** 3

**Review:**

Overall:
The paper considers a problem of multi-class segmentation in an unsupervised setting. The authors propose to use adversarial training to train a model without ground-truth labels. The presented approach is applied to cardiac CT and MRI scans. I like the paper very much and in my opinion the idea is novel and interesting also for other applications. It is one of best papers about adversarial training I read recently.

Strengths:
+ The paper is well written and easy to follow.
+ The proposed approach (adversarial training and an unsupervised loss) is well-thought and very suitable in the considered problem.
+ The proposed approach is novel and could be used in other applications.
+ The experiment is very well described and all results are convincing.

Remarks:
* Minor
- I suggest to remove a reference ([1]) from the abstract.
- How the variance in c_4(X,Z) is calculated?
- What are training and testing wall-clock times?

**Special Issue:**

Definitely

---

### Comment · ~Bram_van_Ginneken1 · 2018-05-18
**Selection for longlist for special issue Medical Image Analysis**

Dear authors,

Congratulations on your acceptance to MIDL! We have selected your paper on the longlist for the Medical Image Analysis Special Issue. Please read this page:
https://midl.amsterdam/special-issue-in-medical-image-analysis/
Please answer the three questions that are listed on that page about your interest in submitting to the special issue, potential overlap with other publications, and related publications.

You can post your answer here directly below on openreview.net, or mail me directly at bram.vanginneken@radboudumc.nl.

Best regards, Bram

---

### Decision · Program_Chairs · 2018-05-15
**Paper111 Acceptance Decision**

Oral